# Training-Free Activation Sparsity in Large Language Models

**James Liu**[1,2]* **Pragaash Ponnusamy**[2] **Tianle Cai**[3] **Han Guo**[1] **Yoon Kim**[1] **Ben Athiwaratkun**[2]

[1] Massachusetts Institute of Technology  [2] Together AI  [3] Princeton University

 https://github.com/FasterDecoding/TEAL

## Abstract

Activation sparsity can enable practical inference speedups in large language models (LLMs) by reducing the compute and memory-movement required for matrix multiplications during the forward pass. However, existing methods face limitations that inhibit widespread adoption. Some approaches are tailored towards older models with ReLU-based sparsity, while others require extensive continued pre-training on up to hundreds of billions of tokens. This paper describes TEAL (**T**raining-Fre**e A**ctivation Sparsity in **LL**Ms), a simple training-free method that applies magnitude-based activation sparsity to hidden states throughout the entire model. TEAL achieves 40-50% model-wide sparsity with minimal performance degradation across Llama-2, Llama-3, and Mistral families, with sizes varying from 7B to 70B. We improve existing sparse kernels and demonstrate wall-clock decoding speed-ups of up to 1.53× and 1.8× at 40% and 50% model-wide sparsity. TEAL is compatible with weight quantization, enabling further efficiency gains.

## 1 Introduction

Large language models (LLMs) demonstrate that scaling in both parameter count and training data leads to capabilities that are useful for addressing a variety of downstream tasks (Brown et al., 2020). However, the large number of parameters in modern LLMs can lead to substantial challenges during inference. In typical small-batch deployment settings, autoregressive inference is *memory-bound*, i.e., bottlenecked by the speed at which the parameters can be moved from off-chip to on-chip memory. This is in contrast to LLM training and prefill inference, which is generally *compute-bound*, i.e., bottlenecked by the speed at which computation can performed. A core strategy for overcoming this *memory wall* (Gholami et al., 2024) is through weight quantization (Frantar et al., 2022; Shao et al., 2023; Yuan et al., 2023; Lin et al., 2024; Dettmers et al., 2023c; Tseng et al., 2024; Egiazarian et al., 2024; Liu et al., 2024) and sparsification (Wang et al., 2019; Frantar & Alistarh, 2023; Xia et al., 2023; Ma et al., 2023), which can lead to practical speed-ups when coupled with specialized kernels that move the weights from off-chip to on-chip memory in quantized/sparse formats (Kim et al., 2023; Dettmers et al., 2023b; Frantar et al., 2024; Wang et al., 2024b; Xia et al., 2024; Guo et al., 2024).

The above methods directly compress a model's weights and apply the same (quantized/sparse) matrix to all inputs. Activation sparsity (Chen et al., 2023; Raihan & Aamodt, 2020; Kurtz et al., 2020) is an alternative method which enforces *input-dependent* structure on the weight matrices by leveraging (or inducing) sparsity in the hidden states. Since the weight channels corresponding to zero-valued activations are not used in computation, speed-up can be realized by selectively omitting these weights during memory transfer, which is possible due to the hardware-friendly channel-wise sparsity pattern. In older LLMs, activation sparsity is largely made possible by the high natural sparsity (around 95%) in the intermediate states of the MLP blocks in ReLU-based Transformer models (Li et al., 2023). Based on this, Liu et al. (2023) propose *DejaVu*, which learns a small auxiliary model that predicts the contextual activation sparsity patterns of future layers, and realize a 2× wall-clock speed-up on OPT-175B (Zhang et al., 2022a). Because the hidden state is extremely sparse, the less expressive auxiliary model can afford to overestimate non-zero activations while maintaining accuracy and efficiency (e.g., 20% predicted vs. 5% actual non-zero entries).

---

*Correspondence to `jamesll@mit.edu`. Work done during an internship at Together AI.

However, modern LLMs have largely moved away from ReLU-based feedforward layers due to their worse performance compared to variants like SwiGLU (Shazeer, 2020). In these models the activations are no longer naturally sparse, making it difficult to apply methods like DejaVu. And while recent works have found that replacing SiLU with ReLU in the MLP blocks and performing continued pre-training can "recover" models that exhibit high activation sparsity (thus making older methods applicable) (Mirzadeh et al., 2023; Song et al., 2024a;b), such methods require training on up to hundreds of billions of tokens.

This work describes TEAL (**T**raining-Fre**e** **A**ctivation Sparsity in **L**LMs), a simple, training-free approach that applies activation sparsity based on magnitude pruning. TEAL is

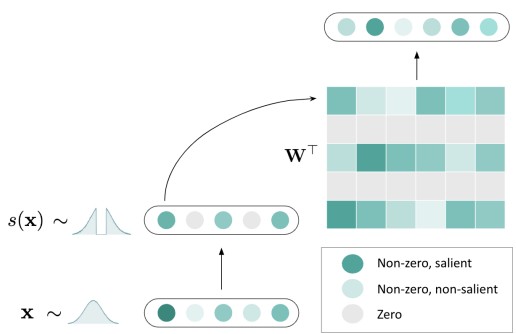

Figure 1: **Overview of TEAL**. During decoding, TEAL thresholds low-magnitude activation entries to zero, which obviates the need to move the associated weight channels onto the registers, thus enabling wall-clock speed-ups.

based on the observation that distributional shapes in LLaMA-architecture LLMs are zero-mean unimodal. By pruning low-magnitude, non-salient activations, we achieve 40-50% model-wide (input-dependent) sparsity, in contrast to prior work which only achieves sparsity in portions of the model (Lee et al., 2024b). We realize wall-clock speed-ups of up to $1.53\times$ and $1.8\times$ at 40% and 50% sparsity respectively through specialized kernels, and further demonstrate compatibility with weight quantization.

## 2 RELATED WORK

Conditional computation (Bengio, 2013; Bengio et al., 2016) alleviates the burden of training and serving by selectively activating parts of a model. Shazeer et al. (2017) propose Mixture-of-Experts (MoE) in language models, applying conditional computation to feed forward networks. Mixture-of-Experts models decouple parameter count with computational footprint (Fedus et al., 2022), and demonstrate superior scaling laws compared to dense baselines (Clark et al., 2022). Dense models can also be converted into MoE models after pre-training (Zhang et al., 2022b; Szatkowski et al., 2024; Zheng et al., 2024).

Activation sparsity occurs when a significant portion of a model's hidden states contain zero-valued entries, and can be seen as an instance of conditional computaton. Activation sparsity is known to naturally emerge in the intermediate states of ReLU-based MLPs (Li et al., 2023). Liu et al. (2023) leverage activation sparsity to accelerate LLM inference by avoiding the transfer of weight channels associated with zero-valued entries to GPU registers. Song et al. (2023) and Alizadeh et al. (2024) extend activation sparsity to CPU offloading, reducing weight transfer from CPU to GPU memory. However, newer architectures typically make use of non-ReLU-based MLPs (e.g., SwiGLU, Shazeer, 2020), making these off-the-shelf methods difficult to use in practice.

Recent work has thus focused on reintroducing activation sparsity in newer architectures. Mirzadeh et al. (2023) replace SiLU or GeLU activation functions with ReLU, followed by continued pretraining on hundreds of billions of tokens. Zhang et al. (2024b) experiment with different activations and find Squared ReLU (So et al., 2022) to be the most effective replacement. Song et al. (2024b) and Song et al. (2024a) introduce techniques such as activation regularization to push sparsity even higher in adapted models. Wang et al. (2024a) combine magnitude pruning with Squared ReLU and quantized activations, and establish scaling laws for sparsely activated LLMs during pretraining.

Lee et al. (2024a) propose *CATS*, and realize training-free activation sparsity on SwiGLU based LLMs by applying magnitude pruning on the output of $\mathbf{W}_{\text{gate}}$, with the intuition that in the training-free setting, ReLU-based methods suboptimally zero out nontrivial negative values but keep positive values with lower magnitude intact. They achieve up to 50% sparsity in $\mathbf{W}_{\text{up}}$ and $\mathbf{W}_{\text{down}}$ for Mistral and Llama-2-7B. However, other matrices including $\mathbf{W}_{\text{gate}}$ and $\mathbf{W}_{\text{q,k,v,o}}$ are computed densely, resulting in lower model-wide sparsity (roughly 25%), whereas we target every matrix in the model. We refer the reader to Appendix A.3 for formal definitions of the weight matrices and their interactions within each Transformer block.

## 3   BACKGROUND: ACTIVATION SPARSITY IN NEURAL NETWORKS

The activation sparsity of a hidden state $\mathbf{x}$ is defined as the proportion of zero-valued entries, which can interact with the model in two ways. The first is *input sparsity*: when computing $\mathbf{y} = \mathbf{x}\mathbf{W}^{\top}$ for $\mathbf{x} \in \mathbb{R}^m, \mathbf{W} \in \mathbb{R}^{n \times m}$, the columns $\mathbf{W}_{:,i}$ corresponding to zero-valued entries $\mathbf{x}_i$ are unused. The second is *output sparsity*: when computing $\mathbf{y} = \mathbf{s} \odot (\mathbf{x}\mathbf{W}^{\top})$ for the aforementioned parameters and mask $\mathbf{s} \in \mathbb{R}^n$, the rows $\mathbf{W}_{i,:}$ corresponding to zero-valued entries $\mathbf{s}_i$ are unused. CATS makes use of output sparsity on GLU variants, treating $\mathbf{s} = \mathrm{sparsify}(\sigma(\mathbf{x}\mathbf{W}_{\mathrm{gate}}^{\top}))$ as the mask and applying output sparsity on $\mathbf{x}\mathbf{W}_{\mathrm{up}}^{\top}$, with the intuition that $\sigma(\cdot)$ serves as a gating mechanism. Interestingly, we find in Section 5.4.1 that input sparsity is still preferable in the training-free case for SwiGLU.

In LLMs, the computation $\mathbf{x}\mathbf{W}^{\top}$ is memory-bound in the decoding phase due to the high memory footprint of weights, and thus reducing the transfer of unnecessary entries (i.e., rows/columns corresponding to zero-valued activations) can enable speed-ups. However, GPUs are designed to fetch multiple consecutive memory entries in a single access to maximize memory bandwidth. When memory accesses are non-contiguous, as they are when unnecessary entries are scattered, this leads to inefficient use of memory bandwidth. To ensure memory coalescing and contiguous memory access, it is crucial to store weights associated with input sparsity in a *column-major* format, and weights associated with output sparsity in a *row-major* format.

## 4   TEAL: TRAINING-FREE ACTIVATION SPARSITY IN LLMS

### 4.1   MOTIVATING STUDY: DISTRIBUTIONAL PROPERTIES OF ACTIVATIONS IN LLMS

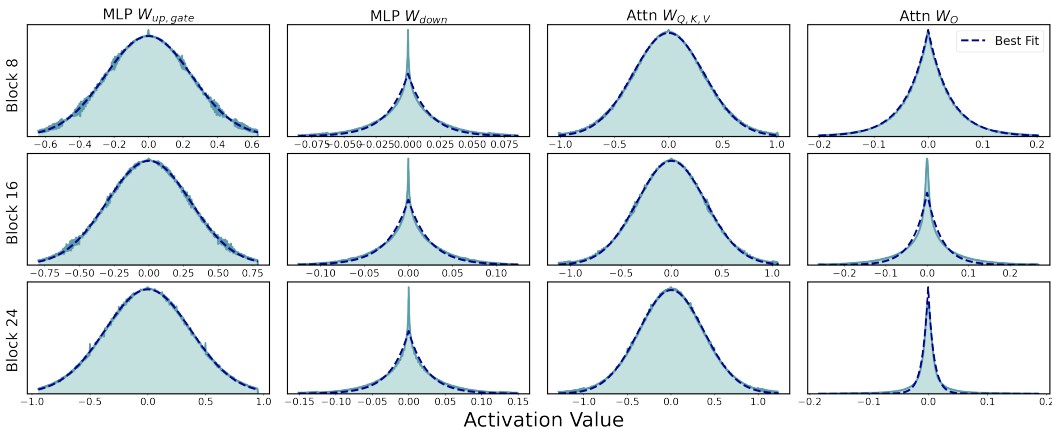

Figure 2: Activation distributions of Llama-3-8B's four hidden states at Blocks 8, 16, and 24. The activations preceding the Attention and MLP blocks typically exhibit Gaussian-like shapes, while intermediate states within these blocks exhibit Laplacian-like shapes. The best-fit Gaussian/Laplace distributions are overlaid in blue.

We perform a preliminary study of the distributional properties of activations of LLMs. We collect activations of Llama-3-8B (Dubey et al., 2024) sampled from C4 (Raffel et al., 2023) at the four hidden states in a Transformer block,[1] and visualize them in Figure 2. As indicated by prior work, some of the activations are heavy-tailed and contain outliers (Dettmers et al., 2022; Xiao et al., 2022; Wei et al., 2022; Nrusimha et al., 2024). The hidden states are moreover zero-mean unimodal, and qualitatively fall into two distinctly shaped distributions. The hidden states before the Attention and the MLP layers tend to be Gaussian-like, while the hidden states in the intermediate of such layers tend to be Laplacian-like. The concentration of the activations around zero motivates our magnitude-based activation pruning approach.

**Remark.** We do not attempt to explain why these distributions are shaped the way they are, nor do we give the theoretical underpinnings of why activation sparsity works. However, we make a few general observations. LLM weights are typically Gaussian (Dettmers et al., 2023a), and multiplying

---

[1]Throughout, we use "block" to refer to an entire Transformer block consisting of the seven matrices and "layer" to refer to an individual layer (corresponding to a single matrix) within the Transformer block.

an independent isotropic Gaussian vector with an independent Gaussian matrix follows a multivariate generalized Laplace distribution Mattei (2017) (the weights and activations are clearly not independent in practice). Attention is a data-dependent linear operator (Poli et al., 2023) which may have similar properties. Distributions may be zero-mean due to layer normalization (Ba et al., 2016). We further derive the expected error induced by pruning low-magnitude activations in Appendix A.1, under a more restrictive assumption that weights and activations are independent Gaussians.

## 4.2 TEAL

The above analysis motivates our simple approach for activation sparsity based on magnitude pruning. While small-magnitude activations could still have a large effect on the output if the norms of corresponding channels of the weight matrix are large, we find that magnitude-based pruning is empirically effective. We first define a sparsification function for an activation vector as follows:

**Definition 1.** *For a random vector* $\tilde{\mathbf{x}} = (\tilde{x}_1, \ldots, \tilde{x}_n)$ *and sparsity level* $p \in [0, 1]$*, define the threshold* $t_p$ *as*

$$\frac{1}{n} \sum_{i=1}^{n} \mathbb{P}(|\tilde{x}_i| \leq t_p) = p.$$

*The sparsification function* $s_{t_p} : \mathbb{R}^n \to \mathbb{R}^n$ *is defined as:*

$$s_{t_p}(\mathbf{x}) = (s_{t_p}(x_1), \ldots, s_{t_p}(x_n))$$

*where* $\mathbf{x}$ *is a realization of* $\tilde{\mathbf{x}}$*, and for each component:*

$$s_{t_p}(x_i) = \begin{cases} 0 & \text{if } |x_i| \leq t_p \\ x_i & \text{otherwise} \end{cases}$$

In practice we estimate $t_p$ using an empirical distribution constructed offline using activations from generic text. The sparsity level $p$ is characterized entirely by threshold $t_{p_i}$, which is useful in both implementation and kernel design (Section 4.4).

Let $\mathcal{W}$ be the set of matrices in the MLP and Attention blocks of a model, and further let $N = |\mathcal{W}|$. We define a model-level sparsification configuration as $\mathbf{p} = (p_1, ..., p_N)$, where each $p_i \in [0, 1]$ represents the sparsity level for the corresponding matrix $\mathbf{W}_i$. For each matrix $\mathbf{W}_i \in \mathcal{W}$, we define its layer-level sparsified forward pass as:

$$\hat{\mathbf{Y}} = s_{t_{p_i}}(\mathbf{x})\mathbf{W}_i^\top$$

for input $\mathbf{x}$ and magnitude-based sparsification function $s_{t_{p_i}}(\cdot)$ as defined in Definition 1. We apply this sparsified forward pass to all $N$ matrices to obtain the model-level sparsified forward pass.

## 4.3 BLOCK-WISE GREEDY OPTIMIZATION

How should we find the optimal $\mathbf{p}$? We initially tried a gradient-based approach to learning the thresholds based on the straight through estimator (Bengio et al., 2013), but encountered optimization issues. We instead used a simple greedy approach illustrated in Algorithm 1, which was found to be effective.

For each Transformer block, we seek to minimize the block-wise $\ell_2$ activation error subject to a block-level sparsity constraint. Each Transformer block consists of seven matrices: $\mathbf{W}_q, \mathbf{W}_k, \mathbf{W}_v, \mathbf{W}_o, \mathbf{W}_{gate}, \mathbf{W}_{up}, \mathbf{W}_{down}$. Algorithm 1 initializes the sparsity levels of all layers to zero, and attempts to increment the sparsity level of each layer by an amount inversely proportional to its memory footprint. The layer with the lowest $\ell_2$ activation error is incremented, and the block-level sparsity plus associated layer-level sparsities

---

**Algorithm 1** Block-wise Greedy Optimization

**Input:** Block $B$, base step size $\alpha$,
    input $\mathbf{X} \in \mathbb{R}^{B \times seq \times d}$, $n$ matrices
  # Record size (memory footprint) of matrices
  $f_i \leftarrow \text{size}(\mathbf{W}_i)$ for $i = 1, \ldots, n$
  $F \leftarrow \sum_{i=1}^{n} f_i$ # Find size of block
  # Initialize block and layer sparsities to zero
  $\mathbf{p} \leftarrow \mathbf{0}_n, P \leftarrow 0$
  $\mathbf{Y}_{gt} \leftarrow B(\mathbf{X})$ # Forward pass through block $B$ to find ground truth output
  **while** $P < 1$ **do**
    **for** $i = 1$ to $n$ **do**
      $\delta_i \leftarrow \alpha \cdot (F/f_i)$
      # Error if we further sparsify this layer
      $p_i \mathrel{+}= \delta_i$
      $\hat{\mathbf{Y}}_i \leftarrow L(\mathbf{X}, p_i')$
      $E_i \leftarrow \|\mathbf{Y}_{gt} - \hat{\mathbf{Y}}_i\|_2$
      $p_i \mathrel{-}= \delta_i$
    **end for**
    $j \leftarrow \arg\min_i E_i$
    $p_j \mathrel{+}= \delta_j$ # Increment layer with lowest error
    $P \leftarrow \sum_{i=1}^{n} (p_i \cdot f_i)/F$
    Record $\mathbf{p}$, $P$
  **end while**

---

are recorded. We assign the same block-level sparsity level to every Transformer block; therefore, all blocks have the same target sparsity level, but the individual layer-level sparsities could be different across different blocks.

**Cost.** We describe the cost of our method. The time complexity is $\mathcal{O}(\frac{Mn^2}{\alpha})$ forward passes, where $M$ is the number of samples, $n$ is the number of matrices, and $\alpha$ is the average step size. In practice, we use length 2048 samples and $M, n, \alpha = 10, 7, 0.05$. The resulting cost over all blocks is therefore $10 \cdot 7^2 \cdot \frac{1}{0.05} = 9800$ forward passes, which is less than one GPU-hour on an A100 for Llama-3-8B. It consumes minimal device memory due to its being block-wise and requiring no backpropagation.

### 4.4 Hardware Aware Acceleration

We develop a specialized sparse GEMV kernel to achieve practical speed-ups, building on the Triton-based (Tillet et al., 2019) kernel introduced by DejaVu (Liu et al., 2023). This kernel takes in an input $\mathbf{x}$, boolean sparsity mask $\mathbf{s}$ and matrix $\mathbf{W}$, and returns $(\mathbf{x} \odot \mathbf{s})\mathbf{W}^\top$. Wall-clock speed-up is realized in three ways: (1) $\mathbf{W}$ is stored in column major format for optimal memory coalescing; (2) Columns $\mathbf{W}_{:,i}$ are selectively loaded based on the truth value of $\mathbf{s}_i$; (3) SplitK work decomposition is used, enabling finer-grained parallelism across thread blocks, combining partial results through atomic adds.

Our kernel makes the following improvements on top of the original kernel: (1) We fuse the mask creation process, as $\mathbf{s} = \mathbf{x}[|\mathbf{x}| > t_p]$ is entirely characterized by $\mathbf{x}$ and $t_p$ in TEAL; (2) We accumulate along the outer SplitK dimension in FP16 (keeping the inner in-register accumulation in FP32), as writing to global memory in FP32 results in significant traffic; (3) We specify an eviction policy in PTX, prioritizing cache retention for activations which are reused across multiple thread blocks, and deprioritizing weights which are block-specific. This guarantees that activations are persistent in L2 cache.

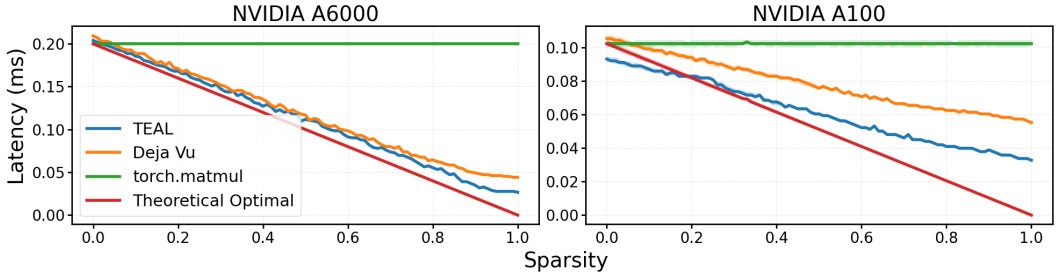

Figure 3: Latency vs. sparsity for matrix-vector multiplication (1x4096 × 4096x14336), comparing TEAL to Deja Vu. 'Theoretical Optimal' shows the latency reduction for torch.matmul assuming perfect linear scaling with sparsity.

Figure 3 shows a small speed-up on A6000, and a larger speed-up on A100 over the DejaVu kernel. Note that torch.matmul is not the strongest baseline in small batch settings (Hong et al., 2024), which is why our kernel is faster at 0% sparsity for A100. We use a stronger baseline for end-to-end evaluations (Section 5.2). The larger speed-up on A100 can be attributed to its higher memory bandwidth, which amplifies the impact of reducing other overhead factors. These overhead improvements become increasingly important as memory bandwidth across device tiers improves over time, particularly for quantized models and in latency-sensitive or resource-constrained applications.

## 5 Results

**Models and Datasets.** We evaluate TEAL on the Mistral (Jiang et al., 2023), Llama-2 (Touvron et al., 2023), and Llama-3 (Dubey et al., 2024) families. We measure the performance of sparsified models on language modeling using the WikiText (Merity et al., 2016) validation set, and on an aggregate of six downstream tasks using the EleutherAI LM Harness (Gao et al., 2023), including 5-shot MMLU, 25-shot ARC challenge, 10-shot HellaSwag, 5-shot GSM8K, zero-shot PiQA, and zero-shot Winogrande (Hendrycks et al., 2021; Clark et al., 2018; Zellers et al., 2019; Cobbe et al., 2021; Bisk et al., 2019; Sakaguchi et al., 2019). For language modeling, we evaluate all models on the same 128 random samples, using a 2048-token context and 512-token evaluation window.

**Baselines.** We use the block-wise greedily optimized sparsities from Section 4.3 for TEAL, and primarily compare to CATS (Lee et al., 2024a) in its training-free configuration with no finetuning. We report model-level sparsities for all methods.

CATS applies sparsity to MLP parameters, and does not apply sparsity to attention parameters. In particular, CATS sparsifies the output of $\mathbf{W}_{\text{gate}}$, replacing $\text{SiLU}(\mathbf{x}\mathbf{W}_{\text{gate}})$ with $s_{t_p}(\text{SiLU}(\mathbf{x}\mathbf{W}_{\text{gate}}))$ for sparsification function $s_{t_p}$ associated with the distribution of $\text{SiLU}(\mathbf{x}\mathbf{W}_{\text{gate}})$. Overall, CATS sparsifies the intermediate state of the MLP by first performing dense computation on $\mathbf{W}_{\text{gate}}$, enforcing output sparsity on $\mathbf{W}_{\text{up}}$, and then enforcing input sparsity on $\mathbf{W}_{\text{down}}$. This is in contrast with TEAL, which enforces input sparsity on all matrices.

Table 1: Perplexity results. Results between Llama-3 and Llama-2/Mistral are not directly comparable due to differing vocabulary sizes.

| Method / Model | LLaMA-3 | | LLaMA-2 | | | Mistral |
| | **8B** | **70B** | **7B** | **13B** | **70B** | **7B** |
|---|---|---|---|---|---|---|
| Baseline (0%) | 5.87 | 2.93 | 5.07 | 4.50 | 3.12 | 4.92 |
| CATS 25% | 6.78 | 3.64 | 5.52 | 4.99 | 3.42 | 5.87 |
| TEAL 25% | **5.94** | **3.02** | **5.09** | **4.51** | **3.13** | **5.01** |
| CATS 40% | $7.6 \cdot 10^4$ | 96.97 | 43.8 | 53.9 | 171 | $2.8 \cdot 10^4$ |
| TEAL 40% | **6.21** | **3.52** | **5.22** | **4.60** | **3.25** | **5.13** |
| TEAL 50% | 6.67 | 4.30 | 5.43 | 4.76 | 3.50 | 5.31 |
| TEAL 65% | 9.06 | 6.29 | 6.62 | 5.50 | 4.28 | 6.23 |

These methods are decoding solutions primarily, but some of the prefill needs to be sparsified for meaningful evaluation on log-likelihood based tasks (such as language modeling and MMLU). For such tasks we sparsify the second half of prefill along the sequence length dimension. See Section 5.4.3 for a more detailed analysis – most degradation in prefill is associated with the initial tokens, which is likely related to the attention sink phenomenon (Xiao et al., 2024), and we thus need to take care not to sparsify them. We do not sparsify prefill on generation tasks (such as GSM8K).

## 5.1 ACCURACY

**Main Results.** TEAL is performant, as shown in Tables 1 and 2, showcasing near zero degradation at 25%, and minimal degradation at 40% sparsity. At 50% sparsity, Llama-3 variants show slightly more degradation compared to older Llama-2 and Mistral variants which are still fairly performant. This falls in line with prior work showing that quantization techniques are less effective on newer models trained on more tokens (Huang et al., 2024). Most models degrade significantly at 65% sparsity, with the exception of Llama-2-70B which is still reasonably performant. In terms of downstream task results, both of the 70B models are more sparsifiable than their smaller counterparts.

Table 2: Downstream task evaluation results. Reported results are averaged over six tasks. See Appendix A.2 for fine-grained results. We omit CATS 40% as it is degenerate.

| Method / Model | LLaMA-3 | | LLaMA-2 | | | Mistral |
| | **8B** | **70B** | **7B** | **13B** | **70B** | **7B** |
|---|---|---|---|---|---|---|
| Baseline (0%) | 68.07 | 80.41 | 56.50 | 62.01 | 72.65 | 66.96 |
| CATS 25% | 64.15 | 79.25 | 54.60 | 60.48 | 71.93 | 64.25 |
| TEAL 25% | **67.73** | **80.22** | **56.42** | **62.21** | **72.67** | **66.63** |
| TEAL 40% | 66.21 | 79.29 | 55.45 | 61.27 | 72.57 | 65.46 |
| TEAL 50% | 63.42 | 78.26 | 54.26 | 60.41 | 72.02 | 64.16 |
| TEAL 65% | 52.59 | 73.07 | 48.16 | 55.71 | 69.30 | 58.93 |

ReLUfication is degenerate in the training-free setting. TEAL outperforms CATS at both 25% and 40% sparsity, which is mainly due to two factors. First and most importantly, TEAL sparsifies every matrix in the model, not just $\mathbf{W}_{\text{up}}$ and $\mathbf{W}_{\text{down}}$, allowing us to moderate sparsity levels across the model. When applied to Llama-2-7B, CATS sparsifies the intermediate state of MLPs to 56.2% at 25% overall sparsity, and to 89.7% at 40% overall sparsity. TEAL avoids such extreme sparsity in

any single component. Second, our design choice to use input sparsity instead of output sparsity for $\mathbf{W}_{up}$ yields lower error, which we analyze in Section 5.4.1.

Table 3: Single-batch end-to-end inference speed results, measured in tokens per second. We exclude Mistral-7B and Llama-2-70B as they are architecturally similar to Llama-3-8B and 70B. We utilize tensor parallelism for Llama-3-70B: TP2 for A100, and TP4 for A6000.

| GPU | Sparsity | LLaMA-3 | | LLaMA-2 | |
|---|---|---|---|---|---|
| | | **8B** | **70B** | **7B** | **13B** |
| A6000 | Baseline | 45.32 (1.00×) | 15.93 (1.00×) | 50.54 (1.00×) | 26.43 (1.00×) |
| | 0% | 44.49 (0.98×) | 15.57 (0.98×) | 50.06 (0.99×) | 26.25 (0.99×) |
| | 25% | 55.38 (1.22×) | 18.93 (1.19×) | 64.54 (1.28×) | 33.67 (1.27×) |
| | 40% | 64.15 (1.42×) | 20.86 (1.31×) | 77.30 (**1.53**×) | 40.20 (1.52×) |
| | 50% | 73.94 (1.63×) | 23.77 (1.49×) | 89.91 (1.78×) | 47.60 (**1.80**×) |
| A100 | Baseline | 100.79 (1.00×) | 21.85 (1.00×) | 110.15 (1.00×) | 61.01 (1.00×) |
| | 0% | 92.13 (0.91×) | 20.32 (0.93×) | 100.97 (0.92×) | 56.33 (0.92×) |
| | 25% | 112.11 (1.11×) | 25.18 (1.15×) | 126.14 (1.15×) | 70.66 (1.16×) |
| | 40% | 126.24 (1.25×) | 28.78 (1.32×) | 143.85 (1.31×) | 81.90 (1.34×) |
| | 50% | 134.29 (1.33×) | 29.99 (1.37×) | 154.02 (1.40×) | 88.38 (1.45×) |

## 5.2 END-TO-END DECODING SPEED-UP

We benchmark TEAL's end-to-end single-batch decoding latency by integrating it with GPT-Fast (PyTorch, 2024). We enable CUDA graphs and `torch.compile`. Tests use Llama-2 (7B, 13B) and Llama-3 (8B, 70B) models at 0%, 25%, 40%, and 50% uniform sparsities. We use the standard inference benchmarking setup in GPT-Fast, which passes in roughly 5 input tokens and generates at most 200 output tokens. Our GPU power limit settings are 500W and 300W for A100 and A6000 respectively. As shown in Table 3, TEAL achieves significant speed-ups of up to 1.53× and 1.8× at 40% and 50% sparsity respectively. TEAL is slower than the baseline at 0% sparsity on A100 due to `torch.compile` strengthening the `torch.matmul` baseline. This suggests further room for optimization of our kernel. We find lower speedups for Llama-3-8B compared to Llama-2-7B partially due to its larger LM Head, which we do not currently sparsify. We leave the sparsification of LM Head to future work.

## 5.3 COMPATIBILITY WITH QUANTIZATION

We demonstrate compatibility with quantization, which is another promising direction for efficient LLM inference. We consider 8-bit channel-wise RTN, 4-bit AWQ (Lin et al., 2024), and 2/3-bit QuIP# (Tseng et al., 2024), and plot the perplexity of Llama-2-7B on WikiText in Figure 4. The point of sharp perplexity degradation is similar across bitwidths, suggesting that errors from activation sparsity and weight quantization compound somewhat independently. Combining activation sparsity with weight quantization unlocks new regimes with respect to memory transferred to GPU registers, allowing for higher inference speed-up. This requires developing specialized sparse + quantized kernels, which we leave to future work.

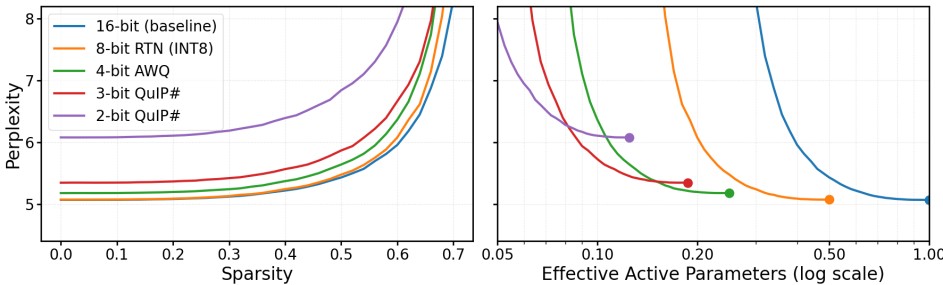

Figure 4: Perplexity vs. sparsity for Llama-2-7B quantized to various bitwidths on WikiText. Left: Performance over sparsity levels. Right: Performance normalized by bitwidth.

### 5.4 ANALYSIS

#### 5.4.1 SHOULD $\mathbf{W}_{\text{UP}}$ HAVE INPUT OR OUTPUT SPARSITY?

TEAL naturally differs from CATS in its treatment of $\mathbf{W}_{\text{up}}$. TEAL uses input sparsity, whereas CATS uses output sparsity with output mask $\mathbf{s} = s_{t_p}(\text{SiLU}(\mathbf{x}\mathbf{W}_{\text{gate}}^{\top}))$, with the intuition that SiLU serves as a gating mechanism. We must choose one treatment over the other due to differing memory format constraints (see Section 3).

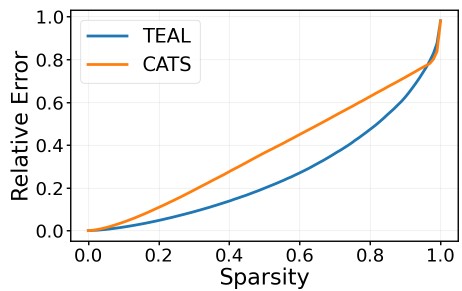
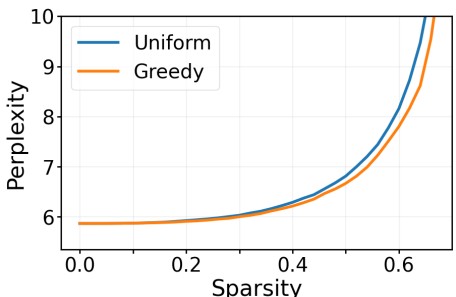

Figure 5: Layer-level activation error for $\mathbf{W}_{\text{up}}$ at Block 16 of Llama-3-8B: TEAL utilizing input sparsity, and CATS utilizing output sparsity.

Figure 6: Perplexity of Llama-3-8B on Wiki-Text under uniform and block-wise greedy sparsity configurations.

We analyze the activation error in the intermediate state of MLPs, assuming $\mathbf{W}_{\text{gate}}$ is computed densely, as it is in CATS. The error associated with TEAL is $||(\mathbf{x} - s_{t_p}(\mathbf{x}))\mathbf{W}_{\text{up}}^{\top} \odot \text{SiLU}(\mathbf{x}\mathbf{W}_{\text{gate}}^{\top})||_2$, and the error associated with CATS is $||\mathbf{x}\mathbf{W}_{\text{up}}^{\top} \odot [\text{SiLU}(\mathbf{x}\mathbf{W}_{\text{gate}}^{\top}) - s'_{t_p}(\text{SiLU}(\mathbf{x}\mathbf{W}_{\text{gate}}^{\top}))]||_2$, where $s_{t_p}(\cdot)$ and $s'_{t_p}(\cdot)$ are sparsification functions associated with $\mathbf{x}$ and $\text{SiLU}(\mathbf{x}\mathbf{W}_{\text{gate}}^{\top})$ respectively. We additionally normalize errors by the norm of the unsparsified product. Figure 5 shows that input sparsity outperforms across all levels. This is because output mask $\mathbf{s}$ has no information regarding the saliency of outputs with respect to $\mathbf{W}_{\text{up}}$, which is relevant since SiLU does not threshold exactly to zero. As a result, larger values of $\mathbf{x}\mathbf{W}_{\text{up}}$ may be unnecessarily pruned.

#### 5.4.2 BLOCK-WISE GREEDY SPARSITIES

We observe in Figure 6 that the block-level greedy method in Section 4.3 outperforms the uniform configuration across all sparsity levels. The resultant sparsities can be used to analyze the workings of modern LLMs. We make two interesting observations about Llama-3-70B, which tend to hold for the other models we analyze.

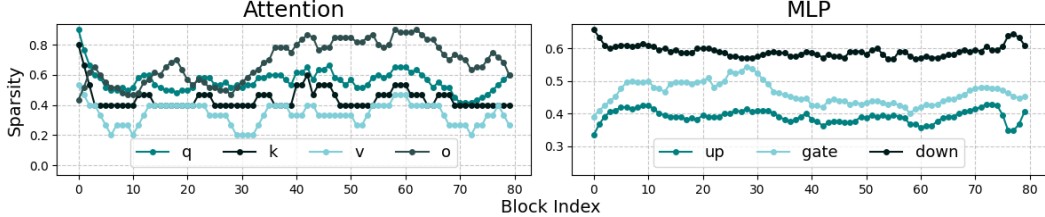

Figure 7: Greedy sparsities for Llama-3-70B at 50% model-level sparsity. Left: Attention parameters. Right: MLP parameters.

**Attention:** We plot sparsities of $\mathbf{W}_{\text{q}}, \mathbf{W}_{\text{k}}, \mathbf{W}_{\text{v}}, \mathbf{W}_{\text{o}}$ at 50% model-level sparsity. $\mathbf{W}_{\text{q}}, \mathbf{W}_{\text{k}}$ exhibit high sparsifiability in Block 0, followed by a sharp decline. $\mathbf{W}_{\text{o}}$'s sparsifiability varies dynamically: it starts at 50-60%, peaks at 80-90% mid-model, then returns to 50-60% in the final blocks. The blocks where $\mathbf{W}_{\text{o}}$ exhibits high sparsifiability seem to align with those of the Attention modules pruned in *FinerCut* (Zhang et al., 2024a), suggesting that the sparsifiability of $\mathbf{W}_{\text{o}}$ may have some correlation to saliency in Attention modules.

**MLP:** We plot sparsities of $\mathbf{W}_{\text{up}}, \mathbf{W}_{\text{gate}}, \mathbf{W}_{\text{down}}$ at 50% model-level sparsity. Across all blocks, $\mathbf{W}_{\text{down}}$ is more sparsifiable than $\mathbf{W}_{\text{gate}}$, which is more sparsifiable than $\mathbf{W}_{\text{up}}$. Intuitively, $\mathbf{W}_{\text{down}}$ is sparsifiable as it corresponds to a Laplacian shaped distribution, which is more densely concentrated around zero than a Gaussian shaped distribution. $\mathbf{W}_{\text{gate}}$ may be more sparsifiable than $\mathbf{W}_{\text{up}}$ due to SiLU decreasing the saliency of negative outputs.

### 5.4.3 PREFILL SPARSIFICATION

We vary the proportion of prefill sparsified (along the sequence length dimension) in Figure 9. Sparsifying the second half of prefill is nearly identical to sparsifying 99% of prefill (all tokens besides the initial tokens). However, more severe degradation occurs when sparsifying the initial tokens. This is due to attention sinks (Xiao et al., 2024), a phenomenon in LLMs where initial tokens are allocated an outsized amount of attention due to the softmax operation. Degradation to keys and values of initial "attention sink" tokens results in more substantial model degradation due to their greater importance (Hooper et al., 2024).

TEAL is a decoding solution so this is typically not an issue, but care must be taken when sparsifying prefill for evaluation on log-likelihood based tasks.

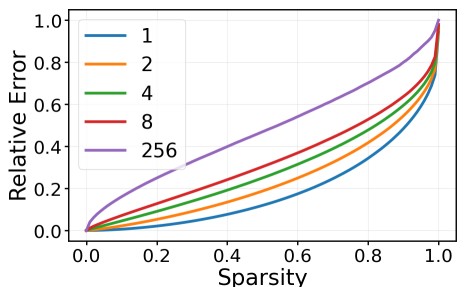
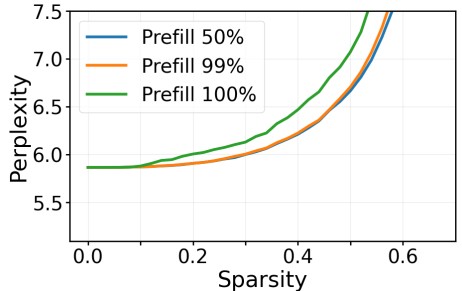

Figure 8: Layer-level activation error for $\mathbf{W}_{\text{down}}$ at Block 16 of Llama-2-7B, at varying batch sizes.

Figure 9: Perplexity of Llama-3-8B on Wiki-Text, varying the proportion of prefill sparsified, using greedy sparsity configurations.

### 5.4.4 BATCHED SPARSIFICATION

We focus on the single-batch case, but it may be valuable to study activation sparsity in batched settings. The key challenge is that different inputs may prefer different sparsity patterns. We need to find a subset of weight columns associated with activations that are relatively low-magnitude for the entire batch.

We propose to sparsify based on the average magnitude of activations across the batch dimension, a natural extension from the single batch case. The resultant sparsification criterion is batch dependent, but is still entirely characterized by a threshold.

As a preliminary analysis, we find the layer-level activation error for $\mathbf{W}_{\text{down}}$ at Block 16 of Llama-2-7B, ablated across batch sizes, in Figure 8. At low batch sizes above 1, $\mathbf{W}_{\text{down}}$ still exhibits substantial sparsity. For example, in the single batch setting, $\mathbf{W}_{\text{down}}$ is assigned roughly 60% sparsity at 50% model-wide sparsity. To have the same error at batch size 4, $\mathbf{W}_{\text{down}}$ is assigned roughly 38% sparsity. As batch size tends to infinity, TEAL can be interpreted as a structured channel-wise pruning algorithm (Zhao et al., 2023), with a simple pruning metric based on activation magnitude.

## 6 APPLICATIONS AND LIMITATIONS

**Applications.** The most immediate application of TEAL is accelerating inference in resource constrained edge settings. These settings are typically single-batch, which is where TEAL realizes the most salient speed-up. Furthermore, TEAL is compatible with quantization (Section 5.3), which is another essential axis of efficiency in this setting.

**Limitations.** TEAL exhibits substantial sparsity in the low-batch setting (Section 5.4.4) but does not scale as well to higher batch sizes, which is a limitation of most activation sparsity work[2]. A way to alleviate this is to push sparsities higher through continued pretraining. While TEAL focuses on the training-free case, we provide many learnings that can aid future work in sparse aware adaptation.

A setting where batched inference is less difficult is in the low-batch setting of Mixture of Experts (Shazeer et al., 2017) based models, as the baseline itself does not scale well due to having to activate more experts and lowering the arithmetic intensity.

## 7 CONCLUSION

We propose TEAL, a simple method that applies magnitude-based activation sparsity to modern LLMs without training, achieving 40-50% model-wide sparsity with minimal degradation. We additionally optimize per-layer sparsity levels, improve existing sparse kernels, and demonstrate compatibility with quantization. We achieve wall-clock speed-ups in single-batch decoding, which is crucial in resource-constrained edge settings. We hope TEAL has impact in real-world applications and enhances our understanding of activation sparsity in LLMs.

ACKNOWLEDGMENTS

We thank Neil Movva, Jue Wang, and Yucheng Lu for helpful comments and discussion.

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

# A APPENDIX

## A.1 DERIVATION OF SPARSIFICATION ERROR

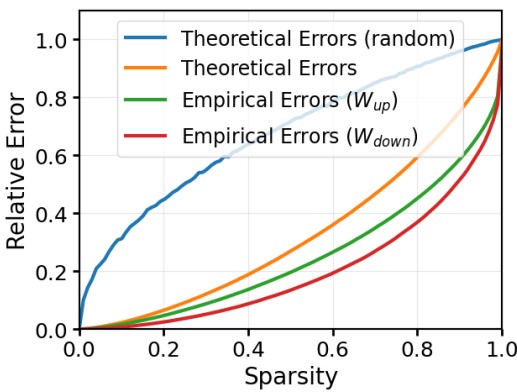

Figure 10: Errors at Block 16 of Llama-3-8B: Gaussian-based theoretical errors from random and magnitude based sparsification, empirical errors from $\mathbf{W}_{\text{up}}$ and $\mathbf{W}_{\text{down}}$.

We derive the error of magnitude-based activation sparsity for the case where $\mathbf{W}$ and $\mathbf{X}$ are independent Gaussian in Theorem A.1. Our error metric is $\frac{\mathbb{E}_{\mathbf{X}}[\|\mathbf{Y}-\hat{\mathbf{Y}}\|_2]}{\mathbb{E}_{\mathbf{X}}[\|\mathbf{Y}\|_2]}$, where $\mathbf{X}$ is the input, $\hat{\mathbf{Y}}$ is the predicted output and $\mathbf{Y}$ is the ground truth output. We plot this error in Figure 10, along with empirical errors on $\mathbf{W}_{\text{up,down}}$ in Block 16 of Llama-3-8B, and the theoretical error obtained from random sparsification.

**Definition A.1.** *For a random vector $\mathbf{X} = (X_1, \ldots, X_n)$ and sparsity level $p \in [0, 1]$, define the threshold $t_p$ as*

$$\frac{1}{n}\sum_{i=1}^{n}\mathbb{P}(|X_i| \leq t_p) = p.$$

*The sparsification function $s_{t_p} : \mathbb{R}^n \to \mathbb{R}^n$ is defined as:*

$$s_{t_p}(\mathbf{X}) = (s_{t_p}(X_1), \ldots, s_{t_p}(X_n))$$

*where for each component:*

$$s_{t_p}(X_i) = \begin{cases} 0 & \text{if } |X_i| \leq t_p \\ X_i & \text{otherwise} \end{cases}$$

**Lemma A.1** (Variance of Scalar Sparsified Error). *For independent random normal variables $X \sim N(0, \sigma_X^2)$, $W \sim N(0, \sigma_W^2)$ and sparsification function $s_{t_p}(\cdot)$, the variance of $(X - s_{t_p}(X))W$ is given by:*

$$\mathrm{Var}((X - s_{t_p}(X))W) = \sigma_X^2 \sigma_W^2 \left[ p - \frac{2t_p}{\sigma_X}\varphi(\frac{t_p}{\sigma_X}) \right]$$

*where $\varphi(t) = \frac{1}{\sqrt{2\pi}}e^{-\frac{1}{2}t^2}$ is the probability density function of the standard normal distribution.*

*Proof.* For $|x| \leq t_p$, $X - s_{t_p}(X)$ follows a truncated normal distribution with lower bound and upper bound given by $-t_p$ and $t_p$ respectively. We thus have:

$$\mathrm{Var}((X - s_{t_p}(X))W) = p\mathrm{Var}(X - s_{t_p}(X) \,\big|\, |X| \leq t_p)\mathrm{Var}(W)$$

$$= \sigma_X^2 p \left[ 1 - \frac{t\varphi(t) - (-t)\varphi(-t)}{\Phi(t) - \Phi(-t)} - \left( \frac{\varphi(t) - \varphi(-t)}{\Phi(t) - \Phi(-t)} \right)^2 \right] \sigma_W^2$$

$$= \sigma_X^2 \sigma_W^2 p \left[ 1 - \frac{2t\varphi(t)}{2\Phi(t) - 1} \right]$$

$$= \sigma_X^2 \sigma_W^2 \left[ p - \frac{2t_p}{\sigma_X}\varphi\left(\frac{t_p}{\sigma_X}\right) \right]$$

where $t = \frac{t_p}{\sigma_X}$, and $\Phi(t) = \frac{1}{2}(1 + \text{erf}(x/\sqrt{2}))$ is the cumulative density function of the standard normal distribution. $\qquad\square$

**Lemma A.2** (Expected $\ell_2$ Norm of Sparsified Matrix-Vector Error). *Let $\mathbf{X} \in \mathbb{R}^m$ be a vector where each $X_i \sim N(0, \sigma_X^2)$, and $\mathbf{W} \in \mathbb{R}^{n \times m}$ be a matrix where each $W_{ji} \sim N(0, \sigma_W^2)$, with all entries independent. For a sparsification function $s_{t_p}(\cdot)$, let $\hat{\mathbf{Y}} = (\mathbf{X} - s_{t_p}(\mathbf{X}))\mathbf{W}^\top$. Then:*

*1) The variance of the $j$-th entry of $\hat{\mathbf{Y}}$ is:*

$$\text{Var}(\hat{Y}_j) = n\sigma_X^2 \sigma_W^2 \left[ p - \frac{2t_p}{\sigma_X}\varphi\left(\frac{t_p}{\sigma_X}\right) \right]$$

*2) The expectation of the $\ell_2$ norm of $\hat{\mathbf{Y}}$ is:*

$$\mathbb{E}[\|\hat{\mathbf{Y}}\|_2] = \sigma_X \sigma_W \sqrt{mn\left[ p - \frac{2t_p}{\sigma_X}\varphi\left(\frac{t_p}{\sigma_X}\right) \right]}$$

*where $t_p$ is the threshold value satisfying $F_{|X|}(t_p) = p$, and $\varphi(t) = \frac{1}{\sqrt{2\pi}}e^{-\frac{1}{2}t^2}$ is the probability density function of the standard normal distribution.*

*Proof.* For the variance of $\hat{Y}_j$: The $j$-th entry of $\hat{\mathbf{Y}}$ is the sum of $n$ independent products $(X_i - s_{t_p}(X_i))W_{ji}$. Each product has variance $\sigma_X^2 \sigma_W^2 [p - \frac{2t_p}{\sigma_X p}\varphi(\frac{t_p}{\sigma_X})]$. Since variances of independent terms add, we multiply this by $n$ to get the result.

For the expectation of $\|\hat{\mathbf{Y}}\|_2$: We first show $\text{Cov}(Y_j, Y_k) = 0$ for $j \neq k$:

$$\mathbb{E}[\hat{Y}_j \hat{Y}_k] = \mathbb{E}\left[ \sum_{i=1}^{n}(X_i - s_{t_p}(X_i))^2 W_{ji}W_{ki} \right] = 0$$

$\mathbb{E}[W_{ji}W_{ki}] = 0$ for $j \neq k$ due to independence and zero mean. $\hat{Y}_j$ and $\hat{Y}_k$ are uncorrelated for $j \neq k$ and are thus independent. Therefore:

$$\mathbb{E}[\|\hat{\mathbf{Y}}\|^2] = \mathbb{E}\left[ \sum_{j=1}^{m} \hat{Y}_j^2 \right] = \sum_{j=1}^{m} \mathbb{E}[\hat{Y}_j^2] = \sum_{j=1}^{m} \text{Var}(\hat{Y}_j)$$

Substituting the variance from part 1, summing over $m$ components, and taking the square-root completes the proof. $\qquad\square$

**Theorem A.1** (Distributional Relative Error). *Let $\mathbf{X} \in \mathbb{R}^m$ and $\mathbf{W} \in \mathbb{R}^{n \times m}$ with elements independently drawn from $N(0, \sigma_X^2)$ and $N(0, \sigma_W^2)$ respectively. For a sparsification function $s_{t_p}(\cdot)$, define $\hat{\mathbf{Y}} = s_{t_p}(\mathbf{X})\mathbf{W}^\top$ and $\mathbf{Y} = \mathbf{X}\mathbf{W}^T$. The distributional relative error is given by:*

$$\frac{\mathbb{E}_{\mathbf{X}}[\|\mathbf{Y} - \hat{\mathbf{Y}}\|_2]}{\mathbb{E}_{\mathbf{X}}[\|\mathbf{Y}\|_2]} = \sqrt{p - \frac{2t_p}{\sigma_X}\varphi\left(\frac{t_p}{\sigma_X}\right)}$$

*where $\varphi(t) = \frac{1}{\sqrt{2\pi}}e^{-\frac{1}{2}t^2}$ is the standard normal probabilty density function.*

*Proof.* From the previous theorem, we have $\mathbb{E}_{\mathbf{X}}[\|\mathbf{Y} - \hat{\mathbf{Y}}\|_2] = \sigma_X \sigma_W \sqrt{mn\left[p - \frac{2t_p}{\sigma_X}\varphi(\frac{t_p}{\sigma_X})\right]}$.

For the unsparsified case, we have $\mathbb{E}_{\mathbf{X}}[\|\mathbf{Y}\|_2] = \sigma_X \sigma_W \sqrt{mn}$. Dividing these expectations yields the result. $\qquad\square$

## A.2 Full Downstream Task Results

We provide the full downstream task results for all evaluated models. For Llama-3-8B in Table 4, we also provide results obtained from the uniform sparsity configuration, showing that the greedy sparsity configuration outperforms across the board. For CATS, we additionally provide the sparsity of the hidden state in the intermediate of the MLP blocks.

Table 4: Full downstream task results for Llama-3-8B.

| Method | Sparsity | MMLU | ARC | HellaSwag | GSM8K | PiQA | WinoGrande | Average |
|---|---|---|---|---|---|---|---|---|
| Baseline | 0 | 65.08 | 57.68 | 82.20 | 49.81 | 80.79 | 72.85 | 68.07 |
| Uniform | 25 | 64.48 | 57.34 | 81.63 | 48.52 | 79.92 | 73.88 | 67.63 |
| | 40 | 62.35 | 54.86 | 79.98 | 43.59 | 79.11 | 72.22 | 65.35 |
| | 50 | 59.07 | 53.50 | 77.60 | 36.47 | 79.22 | 70.17 | 62.67 |
| | 65 | 43.48 | 42.58 | 65.81 | 16.22 | 75.84 | 65.43 | 51.56 |
| Greedy | 25 | 64.61 | 57.17 | 81.77 | 48.90 | 80.47 | 73.48 | 67.73 |
| | 40 | 62.69 | 57.08 | 80.42 | 43.82 | 80.47 | 72.77 | 66.21 |
| | 50 | 59.68 | 53.84 | 78.44 | 38.06 | 78.94 | 71.59 | 63.42 |
| | 65 | 44.54 | 43.86 | 68.85 | 17.51 | 77.15 | 66.06 | 52.99 |
| CATS | 25 (46.45) | 61.18 | 54.61 | 80.20 | 39.88 | 79.76 | 69.30 | 64.15 |

Table 5: Full downstream task results for Llama-3-70B.

| Method | Sparsity | MMLU | ARC | HellaSwag | GSM8K | PiQA | WinoGrande | Average |
|---|---|---|---|---|---|---|---|---|
| Baseline | 0 | 78.70 | 69.71 | 87.94 | 81.12 | 84.55 | 80.43 | 80.41 |
| Greedy | 25 | 78.44 | 69.37 | 87.83 | 80.36 | 84.55 | 80.74 | 80.22 |
| | 40 | 77.92 | 68.52 | 87.35 | 78.77 | 83.79 | 79.40 | 79.29 |
| | 50 | 76.48 | 67.75 | 86.73 | 78.17 | 83.08 | 77.35 | 78.26 |
| | 65 | 71.39 | 63.31 | 83.71 | 64.44 | 80.74 | 74.82 | 73.07 |
| CATS | 25 (45.54) | 77.67 | 67.92 | 87.75 | 78.01 | 84.27 | 79.87 | 79.25 |

Table 6: Full downstream task results for Llama-2-7B.

| Method | Sparsity | MMLU | ARC | HellaSwag | GSM8K | PiQA | WinoGrande | Average |
|---|---|---|---|---|---|---|---|---|
| Baseline | 0 | 45.78 | 52.47 | 78.96 | 13.95 | 78.94 | 68.90 | 56.50 |
| Greedy | 25 | 45.34 | 52.56 | 78.66 | 14.25 | 78.78 | 68.90 | 56.42 |
| | 40 | 42.81 | 53.16 | 78.28 | 12.66 | 78.40 | 67.40 | 55.45 |
| | 50 | 40.52 | 52.47 | 76.54 | 10.84 | 77.86 | 67.32 | 54.26 |
| | 65 | 31.63 | 42.83 | 69.64 | 4.62 | 76.55 | 63.69 | 48.16 |
| CATS | 25 (56.2) | 42.05 | 52.39 | 78.20 | 10.24 | 77.64 | 67.09 | 54.60 |

Table 7: Full downstream task results for Llama-2-13B.

| Method | Sparsity | MMLU | ARC | HellaSwag | GSM8K | PiQA | WinoGrande | Average |
|---|---|---|---|---|---|---|---|---|
| Baseline | 0 | 54.76 | 59.39 | 82.18 | 23.05 | 71.98 | 80.69 | 62.01 |
| Greedy | 25 | 54.96 | 59.47 | 82.31 | 23.58 | 72.14 | 80.79 | 62.21 |
| | 40 | 54.15 | 58.02 | 82.11 | 22.44 | 70.72 | 80.20 | 61.27 |
| | 50 | 52.13 | 57.85 | 81.38 | 19.71 | 71.11 | 80.25 | 60.41 |
| | 65 | 44.37 | 53.24 | 76.95 | 12.81 | 68.59 | 78.29 | 55.71 |
| CATS | 25 (56.02) | 52.31 | 57.76 | 82.73 | 20.32 | 70.24 | 79.49 | 60.48 |

Table 8: Full downstream task results for Llama-2-70B.

| Method | Sparsity | MMLU | ARC | HellaSwag | GSM8K | PiQA | WinoGrande | Average |
|---|---|---|---|---|---|---|---|---|
| Baseline | 0 | 68.71 | 67.49 | 87.02 | 52.38 | 82.70 | 77.58 | 72.65 |
| Greedy | 25 | 68.80 | 67.24 | 86.93 | 52.38 | 82.70 | 77.98 | 72.67 |
| | 40 | 67.75 | 66.81 | 86.88 | 53.90 | 82.48 | 77.58 | 72.57 |
| | 50 | 66.79 | 66.72 | 86.38 | 52.69 | 82.59 | 76.95 | 72.02 |
| | 65 | 62.75 | 63.74 | 84.96 | 45.41 | 81.72 | 77.19 | 69.30 |
| CATS | 25 (45.54) | 67.45 | 66.81 | 86.96 | 50.95 | 82.92 | 76.48 | 71.93 |

Table 9: Full downstream task results for Mistral 7B.

| Method | Sparsity | MMLU | ARC | HellaSwag | GSM8K | PiQA | WinoGrande | Average |
|--------|----------|------|-----|-----------|-------|------|------------|---------|
| Baseline | 0 | 62.46 | 61.43 | 83.47 | 38.21 | 82.10 | 74.11 | 66.96 |
| Greedy | 25 | 62.02 | 61.52 | 83.35 | 37.53 | 81.77 | 73.56 | 66.63 |
| | 40 | 61.00 | 60.84 | 82.65 | 33.89 | 81.28 | 73.09 | 65.46 |
| | 50 | 59.02 | 59.90 | 81.38 | 31.62 | 81.28 | 71.74 | 64.16 |
| | 65 | 51.92 | 54.01 | 76.79 | 21.01 | 80.41 | 69.46 | 58.93 |
| CATS | 25 (46.45) | 60.10 | 59.81 | 82.08 | 29.72 | 80.41 | 73.40 | 64.25 |

## A.3 TRANSFORMER ARCHITECTURE OVERVIEW

A Transformer block consists of an attention layer followed by a multilayer perceptron (MLP). Each block contains seven weight matrices that process the input $\mathbf{x} \in \mathbb{R}^d$ in sequence:

**Attention:** In Grouped Query Attention (GQA) (Ainslie et al., 2023), the matrices $\mathbf{W}_q \in \mathbb{R}^{d \times d}$ and $\mathbf{W}_k, \mathbf{W}_v \in \mathbb{R}^{d_h \times d}$ project the input into query, key, and value representations, which are fed into the attention operation. After the attention operation, $\mathbf{W}_o \in \mathbb{R}^{d \times d_h}$ projects the output back to the model dimension.

**MLP:** The SwiGLU (Shazeer, 2020) MLP variant uses three matrices: $\mathbf{W}_{gate}, \mathbf{W}_{up} \in \mathbb{R}^{d_m \times d}$ which project to a higher dimension, and $\mathbf{W}_{down} \in \mathbb{R}^{d \times d_m}$ which projects back to the model dimension. The computation flow is:

$$\text{MLP}(\mathbf{x}) = (\text{SiLU}(\mathbf{x}\mathbf{W}_{gate}^\top) \odot \mathbf{x}\mathbf{W}_{up}^\top)\mathbf{W}_{down}^\top$$

where $\text{SiLU}(\mathbf{x}) = \mathbf{x} \odot \sigma(\mathbf{x})$, $\sigma$ is the sigmoid function, and $\odot$ denotes element-wise multiplication.

## A.4 COMPARISON TO 2:4 WEIGHT SPARSITY

We compare TEAL to MaskLLM (Fang et al., 2024), a state-of-the-art approach to semi-structured 2:4 weight sparsity that learns weight masks through differentiable relaxation. The authors have not released checkpoints as of the time of writing, so we trained our own masks on Llama-3-8B using 2B tokens from C4. We use the greedily optimized sparsities described in Section 4.3 for TEAL, and evaluate both methods on WikiText:

| Model | Perplexity |
|-------|------------|
| Llama-3-8B (baseline) | 5.870 |
| Llama-3-8B + TEAL (50%) | **6.673** |
| Llama-3-8B + MaskLLM (2:4) | 8.532 |
| Llama-3-8B + TEAL + MaskLLM | 9.590 |

We observe that TEAL outperforms MaskLLM, while being training-free. In contrast, MaskLLM is computationally expensive—training on Llama-3-8B requires 2B tokens and approximately 8B frozen parameters plus 12B learnable parameters, which took us roughly 800 H100 hours. The combination of both methods works well, suggesting they are complementary rather than mutually exclusive.

We additionally compare single-batch decoding speed-up on Llama-2-7B using a single A6000 GPU (using MaskLLM's reported numbers from their Table 6):

| Model | Speed-up |
|-------|----------|
| Llama-2-7B + TEAL (50%) | **1.78×** |
| Llama-2-7B + MaskLLM (2:4) | 1.4× |

This comparison may not be fully representative as 2:4 weight sparsity is more performant in high-batch settings and can additionally accelerate the prefill phase.

## A.5 COMPATIBILITY WITH FINE-TUNING

Table 10: Perplexity results with and without fine-tuning on Llama-3-8B.

| Sparsity Level | PPL (No Fine-tuning) | PPL (With Fine-tuning) |
|---|---|---|
| 0% (baseline) | 5.870 | — |
| 50% | 6.673 | 6.622 |
| 60% | 7.827 | 7.515 |
| 70% | 13.39 | 9.927 |
| 90% | $4.141 \cdot 10^5$ | $4.589 \cdot 10^3$ |

While TEAL is primarily designed as a training-free method, it can be further enhanced with fine-tuning. We fine-tune Llama-3-8B using LoRA (Hu et al., 2021) with a rank of 32 (approximately 1% of parameters are trainable) and a learning rate of 0.0002. The model is fine-tuned on 30M tokens from C4. We evaluate on WikiText and use the greedily optimized sparsities described in Section 4.3.

We observe in Table 10 that fine-tuning provides marginal improvements at lower sparsity levels (50-60%). The benefits are more pronounced at higher sparsity levels (70-90%), where fine-tuning helps to recover some of the performance lost due to aggressive sparsification.

