# OpenReview forum: "Training-Free Activation Sparsity in Large Language Models"
_ICLR.cc/2025/Conference — ICLR 2025 Spotlight_

### Official Review · Reviewer_5UWU · 2024-10-30

**Soundness:** 3
**Presentation:** 3
**Contribution:** 3
**Rating:** 8
**Confidence:** 4

**Summary:**

The paper proposes TEAL, a training-free activation sparsification technique for Large Language Model (LLM) acceleration. Modern LLMs adopt more sophisticated activation functions in lieu of ReLU, thus their activations are no longer naturally sparse. TEAL is built upon the observation that the distribution of the inputs to the weight matrices closely follows either a Gaussian or Laplace distribution. The magnitude based sparsification follows, where the threshold is chosen by a block-level greedy optimization process based on the $\ell_2$ error. The whole process does not involve back propagation. The customized GPU kernel attributes the acceleration of LLMs on the NVIDIA A6000 and A100 GPUs.

**Strengths:**

1. The paper is well-written and easy to follow. The challenge is stated clearly and addressed properly.
2. The proposed method achieves meaningful progress on the activation sparsification of the non-ReLU functions in the training-free setting.
3. The sparsified models show actual throughput improvement on the widely used GPUs. Overall, the proposed scheme seems quite practical and can be easily adopted by others.

**Weaknesses:**

A couple of minor concerns are listed below.
1. Although I understand that the paper focuses on the training-free scheme, providing a discussion and/or experimental results on fine-tuned sparse activation models could make the paper stronger and more interesting. Since a couple of baseline papers (CATS and Relufication) adopt training or fine-tuning steps in their pipelines, I am curious about how much improvement a simple fine-tuning method, such as parameter-efficient fine-tuning (e.g., LoRA), using a moderate number of tokens (possibly <100K), can bring to TEAL. If post-sparsification fine-tuning is prohibited due to the attributes of TEAL, this should be discussed in the paper.
2. The standard deviation of the sparsity by TEAL is not reported. Since the sparsity level changes from input to input, providing more statistical information can help in understanding the proposed method better.

**Questions:**

1. Are there any limitations or difficulties caused by TEAL when it comes to training or fine-tuning the sparsified model?
2. Does the Triton kernel need to be rewritten for different GPU architectures?
3. In Table 3, how was the inference speed tested? Was it tested on autoregressive text generation? If so, what are the lengths of the given and the generated text?
4. Does the sparsity level change based on the domain of the text (e.g., code, novel, scientific paper, etc.)?

---

> ### Author Response · Authors · 2024-11-19
>
> We thank the reviewer for the kind review! Please find below our point-by-point response regarding your feedback:
>
> > W1: Although I understand that the paper focuses on the training-free scheme, providing a discussion and/or experimental results on fine-tuned sparse activation models could make the paper stronger and more interesting… If post-sparsification fine-tuning is prohibited due to the attributes of TEAL, this should be discussed in the paper.
>
> > Q1: Are there any limitations or difficulties caused by TEAL when it comes to training or fine-tuning the sparsified model?
>
> We thank the reviewer for the interesting suggestion. TEAL lends itself well to fine-tuning. We fine-tune Llama-3-8B and use a LoRA rank of 32, meaning ~1% of parameters are trainable. We use a learning rate of 0.0002, and fine-tune on 30M tokens (CATS [1] uses around ~160M tokens. 100k tokens is a bit low). We use C4 for the fine-tuning dataset and WikiText for the evaluation dataset. We use greedily optimized sparsity levels.
>
> | Sparsity level | PPL (No fine-tuning) | PPL (With fine-tuning) |
> |----------------|---------------------|----------------------|
> | 0% (baseline) | 5.870 | - |
> | 50% | 6.673 | 6.622 |
> | 60% | 7.827 | 7.515 |
> | 70% | 13.39 | 9.927 |
> | 90% | 4.141 $\cdot 10^5$ | 4.589 $\cdot 10^3$ |
>
> With fine-tuning we find marginal improvements at lower sparsities (50-60%), and more significant improvements at higher sparsities (70-90%).
>
> > The standard deviation of the sparsity by TEAL is not reported. Since the sparsity level changes from input to input, providing more statistical information can help in understanding the proposed method better.
>
> We find statistics on the model-wide sparsity of Llama-3-8B for decoded tokens on the GSM8K dataset. We set the sparsity level to 50% uniform sparsity.
>
> | Statistic | Model-wide Sparsity Level |
> |-----------|--------|
> | Mean | 50.1724% |
> | Std Dev (per token) | 1.8939% |
>
> The variance is nontrivial, but fairly small, and the per-token model-wide sparsities cluster tightly around the mean.
>
> > Does the Triton kernel need to be rewritten for different GPU architectures?
>
> The kernel should support all [triton-compatible GPUs](https://github.com/triton-lang/triton?tab=readme-ov-file#compatibility) out of the box, including NVIDIA GPUs with Compute Capability 8.0+, and AMD GPUs (ROCm 5.2+). The kernel would need to be rewritten for other accelerators --- though, the overall logic in the kernel is fairly architecture agnostic.
>
> > In Table 3, how was the inference speed tested? Was it tested on autoregressive text generation? If so, what are the lengths of the given and the generated text?
>
> We use autoregressive text generation, and use the standard GPT-Fast inference benchmarking setup, which uses ~5 input tokens and 200 max new tokens.
>
> > Does the sparsity level change based on the domain of the text (e.g., code, novel, scientific paper, etc.)?
>
> We set Llama-3-8B to 50% uniform sparsity, and measure the average model-wide sparsity level on a variety of datasets representing a variety of domains: [GSM8K](https://huggingface.co/datasets/openai/gsm8k) for Math, [GitHub Code](https://huggingface.co/datasets/codeparrot/github-code) for Code, [Tiny Textbooks](https://huggingface.co/datasets/nampdn-ai/tiny-textbooks) for Novels, and [Scientific Papers](https://huggingface.co/datasets/zelalt/scientific-papers) for Scientific Papers.
>
> | Dataset | Model-wide Sparsity Level |
> |---------|--------------------------|
> | GSM8K (Math) | 50.122% |
> | Github Code (Code) | 49.809% |
> | Tiny Textbooks (Novels) | 50.300% |
> | Scientific Papers (Scientific Papers) | 50.035% |
>
> Preliminary results imply that code-based domains have slightly lower sparsity levels, although more work would be needed to concretely verify this. Overall, all domains have fairly similar sparsity levels.
>
> [1]: CATS: Contextually-Aware Thresholding for Sparsity in Large Language Models. https://arxiv.org/abs/2404.08763

---

> > ### Comment · Reviewer_5UWU · 2024-11-21
> >
> > Thank you for your response. All of my concerns and questions have been addressed. I believe this is a good paper, and I am inclined to raise my score.

---

> > > ### Author Response · Authors · 2024-11-24
> > >
> > > We thank reviewer 5UWU for their constructive feedback which helped improve our paper a lot.

---

### Official Review · Reviewer_yw38 · 2024-10-30

**Soundness:** 3
**Presentation:** 4
**Contribution:** 3
**Rating:** 8
**Confidence:** 3

**Summary:**

This paper introduces TEAL, a novel magnitude-based pruning approach to achieve activation sparsity in Large Language Models (LLMs). It focuses on LLaMA-architecture LLMs with SiLU-like activation functions, identifying activation sparsity patterns through a greedy search method that does not require continued pre-training. The experiments demonstrate that TEAL achieves comparable performance on both accuracy and inference speed.

**Strengths:**

- The paper provides a high-quality exploration of sparsity patterns in SiLU-like activations, which are widely used in modern LLMs.
- The greedy search-based solution presented is straightforward yet effective.
- The authors demonstrate the accuracy results on several common downstream tasks at various scales LLMs, and the end-to-end throughput experiments show the practical value of TEAL in certain scenarios.

**Weaknesses:**

- While the main focus is on Llama-like LLMs, discussing the applicability of TEAL to other architectures, such as [Mixture-of-Experts](https://arxiv.org/abs/1701.06538) and [Mamba](https://arxiv.org/abs/2312.00752), would enhance the paper's scope.
- The paper lacks a detailed analysis of the memory footprint of the TEAL method, especially in long context scenarios.

**Questions:**

- In Figure 3 (right), why is the theoretical Latency higher than TEAL at small sparsity levels?

---

> ### Author Response · Authors · 2024-11-19
>
> We thank the reviewer for the kind review! Please find below our point-by-point response regarding your feedback:
>
> > While the main focus is on Llama-like LLMs, discussing the applicability of TEAL to other architectures, such as Mixture-of-Experts and Mamba, would enhance the paper's scope.
>
> We thank the reviewer for the interesting suggestion. We are actively working on this, and our intuition is that TEAL will extend nicely to both Mixture-of-Experts and Mamba architectures.
>
> For Mixture-of-Experts models, we can sparsify each expert independently. Concretely, for a SwiGLU-based MoE, each expert will have two hidden states (before $W_{gate}/W_{up}$, before $W_{down}$), each of which we sparsify. We can additionally target the model router if it is insensitive to sparsification.
>
> For Mamba, we can sparsify each of the hidden states that come before a linear layer. We would need to do a more in depth analysis on if any layer is particularly sensitive to sparsification.
>
> > The paper lacks a detailed analysis of the memory footprint of the TEAL method, especially in long context scenarios.
>
> TEAL incurs minimal memory overhead. In practice, we represent each distribution (corresponding to a hidden state) as a histogram with 10,000 bins, and represent each threshold as a float. For Llama-3-8B, roughly
>
> $$num\textunderscore layers * num\textunderscore hidden \textunderscore states \textunderscore per \textunderscore layer * (num \textunderscore bins + 1) * bytes \textunderscore per \textunderscore value = 32 * 4 * 10001 * 4 = 4.88 \text{ MB}$$
>
> is used, which is **0.061% of the model weights’ footprint**. Furthermore, distributions are stored on system RAM (as opposed to GPU DRAM), as the static thresholds can be configured offline. TEAL is just as performant in long context scenarios, as its memory footprint does not scale with sequence length.
>
> > In Figure 3 (right), why is the theoretical Latency higher than TEAL at small sparsity levels?
>
> TEAL is faster than `torch.matmul` as the latter is not the most optimized for skinny matrix-multiplications, which past work also finds [1]. “Theoretical Optimal” is a fit that assumes that `torch.matmul` scales linearly with the sparsity level, so at 0% sparsity the two are equal and are both worse than TEAL. We will clarify this in the revision. We note that we compare against an improved baseline for the end-to-end results in Table 3, which uses TorchInductor to automatically generate faster matrix-multiplication kernels.
>
> [1]: FlashDecoding++: Faster Large Language Model Inference on GPUs. https://arxiv.org/abs/2311.01282

---

### Official Review · Reviewer_rpTk · 2024-11-04

**Soundness:** 4
**Presentation:** 3
**Contribution:** 3
**Rating:** 8
**Confidence:** 3

**Summary:**

The paper proposes a novel sparsification method called TEAL (Training-Free Activation Sparsity in LLMs) which aims to boost up inference speeds of a contemporary LLMs. It zeros out p-th layer-wise percentile of activations closest to 0 in all of the linear projection layers of the Transformer block. Specifically, it is catered to architectures with GeLU, SwiGLU and GeGLU activations (e.g. Llama and Mistral) which do not exhibit large level of natural sparseness, as opposed to older model families (e.g. GPT and OPT) with high sparseness induced by ReLU. The approach doesn't demand extensive fine-tuning or continual pre-training, as the sparsification is applied in online fashion to activations of the model during each forward pass. Only inexpensive greedy search over limited number of hyper-parameters is required to adapt the technique to a model.

The authors conduct experiments with several model families of different sizes, benchmarking speed, perplexity and reasoning abilities of models augmented with different degrees of TEAL activation sparsity. The results indicate that the method substantially increases inference  speeds while retaining good modeling and reasoning abilities of the models.

**Strengths:**

* The paper successively addresses an important and relevant applied  problem: how to speed up inference of **contemporary** and widely used architectures of LLMs such as Llama model family with no or minimal quality loss.

* Suggested solution is conceptually simple, and authors supply custom Triton kernels which makes their method suitable for easy adoption.

* The paper is clear, legible and easy to follow. The results are solid, and the contribution is sound.

* The authors analyze and visualize activation distributions for recent and largely popular Llama-3-8B architecture which gives an intuitive ground for justifying not only their own online training-free activation sparsity, but also various other existing and future algorithms.

* The authors apply sparsity to all weight matrices in a Transformer block as compared to competing approach (CATS, [1]). TEAL induces greater overall sparsity with lesser degradation of performance.

* The method is shown to work well in practice in conjunction with another extremely popular approach to accelerating the inference of LLMs: weight quantization.

[1] Donghyun Lee, Jaeyong Lee, Genghan Zhang, Mo Tiwari, and Azalia Mirhoseini. CATS: Context- aware thresholding for sparsity in large language models. In First Conference on Language Modeling, 2024.

**Weaknesses:**

1. It would be nice to include formal definitions to various matrices (such as $W_{up}$ and $W_{down}$) and layers (MLP, SwiGLU) which are referred to throughout the work.

2. There are no comparisons with structured and semi-structured pruning (e.g. Mask-LLM [1] which was originally evaluated on the same benchmarks with Llama-2 models). Overall, it would be intriguing and instructive to compare differences in quality/ speed, and trade-offs between the TEAL and unstructured/ semi-structured/ structured pruning, and active sparsity inducing algorithms.

3. In particular, comparison with fine-tuning free ReLUfication [2] seems to be a little bit unfair to the spirit of the method as it’s intended by the paper’s authors to be used with fine-tuning or continual pretraining. It would be nice to either completely discard this comparison in Table 1 or to replace it with a stronger baseline, such as full ReLUfication with fine-tuning.

[1] Fang, G., Yin, H., Muralidharan, S., Heinrich, G., Pool, J., Kautz, J., Molchanov, P., & Wang, X. (2024). MaskLLM: Learnable Semi-Structured Sparsity for Large Language Models. ArXiv, abs/2409.17481.
[2] Iman Mirzadeh, Keivan Alizadeh, Sachin Mehta, Carlo C Del Mundo, Oncel Tuzel, Golnoosh Samei, Mohammad Rastegari, and Mehrdad Farajtabar. Relu strikes back: Exploiting activation sparsity in large language models, 2023. URL https://arxiv.org/abs/2310.04564.

**Questions:**

1. Line 107: Which recent work?

2. Could the authors please explicitly explain why activation sparsity error in figure 8 increases with the batch size? Does the algorithm nullify the activations independently across different samples of a batch or activations are zeroed according to some aggregate statistics?

3. It is mentioned in the paper that the efficient low-level Triton kernel is implemented based on Deja vu kernel [1]. Is it possible to apply that kernel directly to the task of sparsifying SwiGLU and GeGLU transformers, and if it is, how it fares against the proposed method in terms of quality?

4. The paper seems to imply but not to explicitly state that the result of (SiLU(xWgate ) ⊙ xWup is also sparsified. Are there any other places where activation sparsity is applied but not mentioned?


[1] Zichang Liu, Jue Wang, Tri Dao, Tianyi Zhou, Binhang Yuan, Zhao Song, Anshumali Shrivastava, Ce Zhang, Yuandong Tian, Christopher Re, and Beidi Chen. Deja vu: Contextual sparsity for efficient llms at inference time, 2023. URL https://arxiv.org/abs/2310.17157.

---

> ### Author Response · Authors · 2024-11-19
>
> We thank the reviewer for the kind review! Please find below our point-by-point response regarding your feedback:
>
> > It would be nice to include formal definitions to various matrices…
>
> We thank the reviewer for the suggestion, and will incorporate this in the next revision.
>
> > There are no comparisons with structured and semi-structured pruning…
>
> We thank the reviewer for the interesting suggestion. Below we compare against MaskLLM [1], a state-of-the-art approach to semi-structured 2:4 weight sparsity that learns weight masks through differentiable relaxation. The authors have not released checkpoints yet, so we trained our own masks on Llama-3-8B, using 2B tokens from C4. We compare against TEAL with greedily optimized sparsities, and evaluate on WikiText.
>
> | Model | PPL |
> |---------|-----|
> | Llama-3-8B (baseline) | 5.870 |
> | Llama-3-8B + TEAL (50%) | 6.673 |
> | Llama-3-8B + MaskLLM (2:4) | 8.532 |
> | Llama-3-8B + TEAL + MaskLLM | 9.590 |
>
> We see that TEAL outperforms MaskLLM. We emphasize that TEAL is **training-free** and MaskLLM is **fairly expensive** (on Llama-3-8B, uses 2B tokens and requires ~8B frozen params, ~12B learnable params. This took us roughly **800 H100 hours**). Interestingly, the combination of the two methods works well, meaning this isn’t an “either-or” situation.
>
> We additionally compare speed on Llama-2-7B on 1xA6000 in single-batch decoding (using MaskLLM’s reported numbers in Table 6):
>
> | Model | Speedup |
> |---------|-----|
> | Llama-2-7B + TEAL (50%) | **1.78x** |
> | Llama-2-7B + MaskLLM (2:4) | 1.4x |
>
> We note this may not be the most fair comparison as 2:4 weight sparsity is more performant in high-batch settings, and can additionally speed up prefill.
>
> > In particular, comparison with fine-tuning free ReLUfication seems to be a little bit unfair…
>
> We initially included these results to emphasize that simply replacing SiLU with ReLU is insufficient with respect to accuracy in the training-free case. However, we agree that comparing to ReLUfication in the training-free setting doesn't properly represent the method as intended, so we will remove this comparison in the next revision.
>
> > Line 107: Which recent work?
>
> This refers to CATS [2] -- we will make the reference more explicit in the next revision.
>
> > Could the authors please explicitly explain why activation sparsity error in figure 8 increases with the batch size? Does the algorithm nullify the activations independently across different samples of a batch or activations are zeroed according to some aggregate statistics?
>
> Activation sparsity results in speedup by avoiding the transfer of unneeded weight channels from off-chip to on-chip memory during decoding. However, this is an issue in batched settings. If a given activation value is extremely large for one client, but relatively small for another client, the weight channel that corresponds to both clients needs to be loaded. This means that on a per-error basis, activation sparsity is less efficient in batched settings.
>
> We propose to zero out activations according to an aggregate statistic. In particular, using the distribution of average activation values over the batch dimension. The resultant sparsification criterion is batch dependent, but is still entirely characterized by a threshold.
>
>
> > It is mentioned in the paper that the efficient low-level Triton kernel is implemented based on Deja vu kernel. Is it possible to apply that kernel directly to the task of sparsifying SwiGLU and GeGLU transformers, and if it is, how it fares against the proposed method in terms of quality?
>
> This depends on the method being used. Using the DejaVu kernel with the original intended method (low rank predictor to get a boolean output sparsity mask, apply input sparsity on $W_{up}/W_{gate}$, apply output sparsity to $W_{down}$) results in substantial degradation as it was originally intended for the 95+% sparsity introduced by ReLU in the intermediate states of MLPs.
>
> Using the DejaVu kernel with TEAL (create boolean sparsity mask from input activations, apply input sparsity to target matrix) is feasible and identical from an accuracy standpoint, but is slower than our kernel (as shown in Figure 3), as we make some additional optimizations (eg. fusing the sparsity mask creation).
>
>
> > The paper seems to imply but not to explicitly state that the result of (SiLU(xWgate ) ⊙ xWup is also sparsified. Are there any other places where activation sparsity is applied but not mentioned?
>
> We will make this more explicit. Activation sparsity is applied to the four hidden states in the model, visualized in Figure 2: Before $W_q/W_k/W_v$, before $W_o$, before $W_{up}/W_{gate}$, and before $W_{down}$.
>
> [1]: MaskLLM: Learnable Semi-Structured Sparsity for Large Language Models. https://arxiv.org/abs/2409.17481
>
> [2]: CATS: Contextually-Aware Thresholding for Sparsity in Large Language Models. https://arxiv.org/abs/2404.08763

---

> > ### Comment · Reviewer_rpTk · 2024-11-26
> >
> > Thanks for the response. A small remark:
> >
> > > "Interestingly, the combination of the two methods works well, meaning this isn’t an “either-or” situation."
> >
> > It follows from the table with comparisons TEAL vs MaskLLM on perplexity, that simultaneous use of both methods leads to worse results than each of them separately.
> >
> > I hope, you will be able to address the high-batch setting in the future work.
> >
> > Overall, I think your work is exciting and contributes positively to the field. I maintain my score.

---

> > > ### Author Response · Authors · 2024-11-27
> > >
> > > We thank reviewer rpTk for their constructive feedback which helped improve our paper a lot.

---

### Official Review · Reviewer_wbxr · 2024-11-04

**Soundness:** 3
**Presentation:** 2
**Contribution:** 2
**Rating:** 6
**Confidence:** 2

**Summary:**

The authors introduce TEAL, a training-free method for inducing activation sparsity in LLMs, achieving 40-50% sparsity with minimal performance loss and up to 1.8× speed-up in decoding. Unlike previous methods, TEAL works across all model layers, supports modern activation functions, and is compatible with quantization for additional efficiency.

**Strengths:**

- TEAL achieves high sparsity (40-50%) without retraining, resulting in faster execution (up to 1.8× speed-up) with minimal performance degradation.

- A range of experiments across various models and model sizes shows that TEAL consistently outperforms other methods, demonstrating its robustness and general applicability

- The proposed method is designed to work seamlessly with newer models that use modern activation functions like SwiGLU, making it applicable to the latest LLM architectures.

**Weaknesses:**

- Although TEAL is adapted for compatibility with newer activation functions like SwiGLU, the core concept of magnitude-based pruning is similar to prior work, which may limit its perceived innovation and contribution.
- The paper does not mention whether the code is available for reproducibility.

**Questions:**

**1. Calibration Set vs. Evaluation Set for Pruning Decisions:**
 - If I understand correctly, the decision to prune specific activations in TEAL is based on a calibration set, with the assumption that this set is representative enough for generalization. However, is it possible that an activation deemed unimportant (and thus pruned) based on the calibration set might behave differently on a different dataset?
 - It appears that C4 is used as the calibration set, while WikiText is used for evaluation. How similar are these datasets in structure and distribution? If you were to repeat the experiment on a different evaluation dataset, would you still observe the same levels of performance degradation?

**2. Pruning Effectiveness in Highly Optimized Newer Models:**

- In line 298, could it be that newer models are using their capacity more fully, meaning that fewer activations are redundant and available for pruning? If so, does this suggest that newer architectures could achieve comparable complexity with fewer parameters, making further compression both unnecessary and less effective?
- In this scenario, how would TEAL still provide an advantage? Could you explain the benefits of TEAL for cases where newer models are already highly optimized?

---

> ### Author Response · Authors · 2024-11-19
>
> We thank the reviewer for the kind review! Please find below our point-by-point response regarding your feedback:
>
> > Although TEAL is adapted for compatibility with newer activation functions like SwiGLU, the core concept of magnitude-based pruning is similar to prior work, which may limit its perceived innovation and contribution.
>
> While magnitude-based pruning on activations is simple, yet effective, we believe there is significant contribution in 1) applying this to **all hidden states** of the model, not just the intermediate state of the MLP, 2) our greedy block-wise optimization method, which exploits the differing importance of layers in the model, and 3) our kernel, which **significantly outperforms** the DejaVu kernel in our setting.
>
> > The paper does not mention whether the code is available for reproducibility.
>
> Indeed, we overlooked mentioning code availability in the paper. The code is currently available as supplementary material, and this will be reflected in the final manuscript.
>
> > 1. Calibration Set vs. Evaluation Set for Pruning Decisions:
>
> We note our choice to use a calibration set with fixed thresholds was inspired by CATS [1], which uses a similar method. We find the perplexity of Llama-3-8B on WikiText at 50% model-wide sparsity. We use the block-wise greedy optimized sparsity levels. We consider two datasets for calibration, C4 and FineWeb.
>
> | Dataset | PPL |
> |---------|-----|
> | Baseline (0% sparsity) | 5.870 |
> | C4 (50% sparsity) | 6.673 |
> | FineWeb (50% sparsity) | 6.681 |
>
> We observe little variation between different calibration sets. Each hidden state only has one degree of freedom (one threshold per hidden state), meaning it’s hard to overfit to a given calibration set.
>
> > 2. Pruning Effectiveness in Highly Optimized Newer Models:
>
> We note that **all model compression methods** are less effective on overtrained models (eg. weight quantization [2,3]), not just TEAL. For example, in 2:4 weight sparsity, MaskLLM [4] achieves 6.72 PPL vs. baseline 5.12 PPL for Llama-2-7B (App. C). However, it only achieves 8.50 PPL vs. baseline 5.76 PPL for Llama-3-8B (App. D). Accordingly, it seems to us that for more overtrained LLMs (at a fixed model size), TEAL would remain viable when applied more moderately, similar to methods like weight quantization.
>
> We note that TEAL becomes **more effective as model size increases**. For example, both Llama-2-70B and Llama-3-70B exhibit substantially less degradation than Llama-2-7B and Llama-3-8B. Q-Sparse [5] finds a similar phenomenon in the pretraining setting, finding that the gap between sparsely activated models and dense baselines decreases as model size increases (Section 3.5). Thus, TEAL becomes more applicable as models increase in size over time.
>
>
> [1]: CATS: Contextually-Aware Thresholding for Sparsity in Large Language Models. https://arxiv.org/abs/2404.08763
>
> [2]: An Empirical Study of LLaMA3 Quantization: From LLMs to MLLMs. https://arxiv.org/abs/2404.14047
>
> [3]: Scaling Laws for Precision. https://arxiv.org/abs/2411.04330
>
> [4]: MaskLLM: Learnable Semi-Structured Sparsity for Large Language Models. https://arxiv.org/abs/2409.17481
>
> [5]: Q-Sparse: All Large Language Models can be Fully Sparsely-Activated. https://arxiv.org/abs/2407.10969

---

> > ### Author Response · Authors · 2024-11-25
> >
> > Dear reviewer wbxr, thank you again for your thoughtful review. As the rebuttal deadline approaches, we just wanted to follow up on whether our response has addressed your concerns. We've additionally attached a new revision, which we elaborate on in the general response. We welcome any additional feedback you may have.

---

> > ### Comment · Reviewer_wbxr · 2024-11-26
> >
> > I appreciate the authors' effort in clarifying the raised issues. I will increase my score accordingly.

---

> > > ### Author Response · Authors · 2024-11-27
> > >
> > > We thank reviewer wbxr for their constructive feedback which helped improve our paper a lot.

---

### Author Response · Authors · 2024-11-24
**General Response**

We appreciate that the reviewers share our excitement about TEAL. Their thoughtful feedback has helped strengthen our paper while highlighting the potential opportunities for activation sparsity to accelerate modern LLM inference. Below we briefly recap our paper and detail our revision addressing the reviewer comments.

### **Recap**
We study activation sparsity in modern LLMs, which forgo the use of the sparsity-inducing ReLU function due to empirical reasons. In particular, we discover that hidden states in modern models are **almost sparse**, and realize significant wall-clock speedups by clipping small, low-magnitude activations to zero. We make further optimizations on both modeling (Section 4.3) and kernel (Section Section 4.4) fronts, and demonstrate compatibility with weight quantization. Our method is simple, cheap, and **training-free**, allowing for widespread adoption.

We are grateful that reviewers consistently acknowledged our method's **effectiveness** and **practical impact**, noting that our work:

* Provides a "**clear, legible and easy to follow**" solution to an "**important and relevant applied problem**", with a "straightforward yet effective" approach and "custom Triton kernels which makes their method **suitable for easy adoption**". (rpTk, yw38, 5UWU)
* "**Consistently outperforms other methods**" while demonstrating effectiveness "on several common downstream tasks at various scales". (wbxr, yw38)

* Provides valuable "analysis and visualization of activation distributions" that gives "**intuitive ground for justifying** not only their own online training-free activation sparsity, but also various other existing and future algorithms". (rpTk)
* Shows **significant speedups** through "**actual throughput improvement** on the widely used GPUs" with "up to 1.8× speed-up". (5UWU, wbxr)

### **Revision**

Thanks to reviewer feedback, we have uploaded a revision with updates highlighted in blue. We group these updates into two main themes:

**Writing clarity + presentation**
* We provide formal definitions for each mentioned weight matrix. (L99-101, App. A.3) (rpTk)
* We replace "Recent work" with "CATS". (L108) (rpTk)
* We clarify that sparsity is applied to all four hidden states of each Transformer block. (L189-191) (rpTk)
* We clarify the meaning of “Theoretical Optimal” and how it relates to `torch.matmul`. (L246-247) (yw38)
* We remove the comparison with ReLUfication. (Table 1) (rpTk)
* We specify the number of input and output tokens used in our latency benchmarking setup. (L343-345) (5UWU)

**Expanded experimental analysis**
* We include a comparison to MaskLLM, a state-of-the-art approach to semi-structured 2:4 weight sparsity. TEAL outperforms in both accuracy and single-batch decoding latency. (App. A.4) (rpTk)
* We demonstrate compatibility with fine-tuning. Fine-tuning provides marginal improvements at lower sparsities (50-60%), and more substantial improvements at higher sparsities (70-90%). (App. A.5) (5UWU)
* We add ablations on the calibration set. We find TEAL is fairly insensitive to choice calibration set. (App. A.6) (wbxr)

We thank all reviewers again for their constructive feedback. We are happy to follow up with any additional questions.

---

### Meta-Review · Area_Chair_HsZv · 2024-12-18

**Metareview:**

Dear Authors,

Thank you for your valuable contribution to the ICLR and the ML community. Your submitted paper has undergone a rigorous review process, and I have carefully read and considered the feedback provided by the reviewers.

This paper proposes a training-free method for achieving activation sparsity in large language models, achieving around 50% sparsity with minimal performance loss. This leads to up to 1.8× speed-up in decoding.  Overall, the paper received mostly positive response from the reviewers (8,8,8,6) scores.

Given this positive assessment, I am willing to recommend the acceptance of your paper for publication.

I would like to remind you to carefully review the reviewer feedback and the resulting discussion. While most reviews were positive, the reviewers have offered valuable suggestions that can further strengthen the quality of the paper. Please take another careful look a the 'weaknesses' section of each reviewer comment. I encourage you to use this feedback to make any necessary improvements and refinements before submitting the final version of your paper.

Once again, thank you for submitting your work to ICLR.

Best,
Area Chair

**Additional Comments On Reviewer Discussion:**

Reviewers pointed out issues in presentation and writing. They also asked for comparisons with some baselines (e.g. structured pruning via Mask-LLM), asked for ablations on the use of calibration data. The reviewers provided a detailed rebuttal, including the comparisons requested by the reviewers. The reviewers found the rebuttal convincing, which led to uniformly positive scores.

---

### Decision · Program_Chairs · 2025-01-22

Accept (Spotlight)